# The Effect of Relative Humidity Dependent Thermal Conductivity on Building Insulation Layer Thickness Optimization

**Wen Yang** [1,2,*], **Guanjie Zhang** [2], **Wenfang He** [1,2] and **Jiaping Liu** [1,2]

[1] State Key Laboratory of Green Building in Western China, Xi'an University of Architecture and Technology, No. 13 Yanta Road, Xi'an 710055, China

[2] School of Architecture, Xi'an University of Architecture and Technology, No. 13 Yanta Road, Xi'an 710055, China

[*] Correspondence: yangwen@xauat.edu.cn; Tel.: +86-187-10-93-5688

**Abstract:** Optimization of insulation layer thickness is a significant factor in energy-efficient building design. Accurate determination of the thickness of the insulation layer will contribute to building energy conservation. In this study, ten typical cities in five thermal zones were selected, and the external thermal insulation of a typical residential building was taken as the research object. Using the degree day method and the economic model of full life cycle cost analysis, the optimal thickness of seven kinds of building insulation materials under absolute dry conditions, the lowest humidity and the highest humidity of the monthly average of the annual daily average were obtained. In addition, the carbon emission, energy saving and recovery period of materials under different working conditions were further obtained through numerical calculation. The results show that the optimum thickness of seven building insulation materials in ten typical cities under three working conditions is 18.21–346.05 mm. Their carbon emission change rate is between −2.7% and 38.6%, energy saving change rate is between −0.4% and 18.4%, and the payback period growth is within 1.5 years. Among them, polystyrene foam is the material least affected by humidity. It is recommended to be the main building insulation material in high humidity areas.

**Keywords:** building insulation materials; thermal conductivity; relative humidity; optimum thickness; carbon emissions

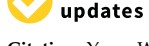



## 1. Introduction

Choosing the right insulation materials for the building envelope and determining the right thickness of insulation are effective methods to achieve energy conservation. Determining the thickness of insulation layers is vital for understanding heat transfers in which thermal conductivity is a crucial factor [1–3]. ASTM standard C168–97 [4] defines the thermal conductivity (W/m·K) as the time rate of steady-state heat flow through the unit area of uniform material, which is caused by the unit temperature gradient perpendicular to the direction of unit area. However, a constant thermal conductivity value is usually substituted in building energy evaluation. The existing specifications in China consider the influence of temperature on the thermal conductivity of building insulation materials but do not mention the influence of humidity on building insulation materials. Lack of specifications can lead to poor thermal performance, thereby increasing building energy consumption and other problems [5].

In the laboratory, in the process of measuring the thermal conductivity of building insulation materials, the moisture content of the materials, as well as the temperature and humidity of the laboratory, will affect the measurement. Thermal insulation materials in buildings differ from their performance under laboratory conditions because they are subjected to a real-world climate during use. See Table 1 for details.

**Table 1.** Research status of building thermal insulation materials affected by ambient temperature and humidity.

| Researcher | Research Contents | Research Results/Innovation |
|---|---|---|
| I. Budaiwi, A. Abdou. [6] | The impact of the k-value change of fibrous insulation materials (i.e., fiberglass) in a typical wall–roof system due to moisture content levels on the thermal and energy performance of a typical residential building under hot–humid climatic conditions is investigated. | Moisture performance is investigated utilizing theoretical longterm hygrothermal performance modeling and simulation techniques. Layer-and time-averaged levels of moisture content in the fibrous insulation are determined, and the corresponding k-value change is evaluated from measured relationships. The impact of the k-value change due to moisture on the building thermal load and cooling energy performance of a residential building is then assessed utilizing detailed building energy simulation software. |
| K.J. Kontoleon, C. Giarma. [7] | This paper investigates the impact of moisture content on the thermal inertia parameters of building material layers. Their consideration is essential to enhance the design of building elements, from a thermal point of view, when exposed to varying moisture content conditions. | Moisture content and relative humidity variations of each analysed layer, as defined by specific moisture storage functions, are shown to interrelate non-linearly with the layer resistor–capacitor circuit section parameters (thermal conductivity and volumetric heat capacity) showing notable consequences on the thermal inertia parameters. |

Although the thermal characteristics of building thermal insulation materials in the real environment are different from those under laboratory conditions, the influence of temperature and humidity on the thermal conductivity measured under laboratory conditions is still the basis of this research. This paper conducts a literature survey on the influence of temperature and humidity on the thermal conductivity, as shown in Table 2.

**Table 2.** Research status of building thermal insulation materials affected by temperature and humidity.

| Researcher | Research Contents | Research Results/Innovation |
|---|---|---|
| I. Budaiwi, A. Abdou, M. Al-Homoud. [8] | They revealed the relationship between the temperature and thermal conductivity of various locally produced insulating materials. | The impact of thermal conductivity variation with temperature on the envelope-induced cooling load for a theoretically modeled building is quantified and discussed. |
| M. Khoukhi, M. Tahat. [9] | This study is to investigate the relationship between the temperature and thermal conductivity of various densities of polystyrene, which is widely used as building insulation material in Oman. | The impact of thermal conductivity variation with temperature on the envelope-induced cooling load for a simple building model is discussed. |
| Hoseini, Atiyeh, Majid Bahrami. [10] | This work presents a comprehensive investigation of aerogel blankets thermal conductivity (k-value) in humid conditions at transient and steady state regimes. Transient plane source (TPS) tests revealed that the k-value of aerogel blankets can increase by up to approximately 15% as the ambient relative humidity (RH) increases from 0% to 90% at 25 °C. | This paper mechanisms affecting the k-value of aerogel blankets as a function of RH and T are investigated. |
| Alvey, Jedediah B, Jignesh Patel, Larry D, Stephenson. [11] | In this paper, the thermal conductivity of several commercially available insulating materials (three kinds of aerogel composite blankets, two kinds of extruded polystyrene foam (XPS) and one kind of foamed polyurethane foam (PUR)) is evaluated as affected by ambient temperature and humidity. | Results indicate that humidity levels play a significant role in PUR performance but not a significant role in XPS performance. The three aerogel composites have mixed results: one has little relationship between moisture content and thermal performance, one is strongly affected by moisture and the remaining is moderately affected by moisture. |

**Table 2.** *Cont.*

| Researcher | Research Contents | Research Results/Innovation |
|---|---|---|
| Nosrati, Roya Hamideh, Umberto Berardi. [12] | In this paper, the change of thermal conductivity of aerogel reinforced materials in the temperature range of −20 °C to +60 °C, and the relative humidity (RH) range of 0% to 95% is studied. | This study shows that compared to the standard testing condition, the maximum increase in the thermal conductivity was 100% under 95% RH, while the greatest temperature-driven increase in the thermal conductivity was 12% at the maximum tested temperature. The humidity-driven changes in the thermal conductivity of aerogel-based products are significantly greater than temperature-driven changes. |

In addition, researchers try to better specify the optimal thickness of insulation for different types of materials when studying factors affecting thermal conductivity. See Table 3 for the research status of optimal thickness. Based on the above research on the status quo, almost all the current research on the optimal thickness ignore the influence of environmental humidity on the thermal conductivity. Thermal conductivity is based on the measured thermal conductivity of the material in the absolute dry state. This will cause the design value of thermal conductivity to be lower than the actual value, resulting in insufficient design of insulation thickness, affecting users' thermal comfort and increasing building energy consumption. Research on the influence of temperature and humidity on building thermal insulation materials shows that the humidity-driven change of thermal conductivity is significantly greater than the temperature-driven change. Therefore, in practical engineering applications, we should first consider the influence of humidity when selecting the thickness of building insulation materials, especially when building walls are exposed to extreme humidity for a long time, so as to better realize building energy conservation and reduce greenhouse gas emissions. Although in some areas with high humidity, the construction of buildings may consider the installation of moisture-proof and moisture-reducing components inside the envelope, in the production, transportation and installation of building materials, the humidity inside the building insulation materials will also reach a dynamic balance with the surrounding environment, so it is important to consider the impact of humidity on the building.

Therefore, the effects of humidity on the thermal conductivity of seven building insulation materials, including polystyrene foam (EPS), extruded polystyrene foam (XPS), polyurethane (PUR), rock wool, centrifugal wool, aerogel-enhanced hollow glass microspheres (HGM) and foamed cement, were obtained through experiments in this paper. Secondly, the heat transfer load of typical insulation structure is analyzed by the degree day method, and then the optimal insulation thickness is calculated using the full life cycle cost analysis (LCCA) economic model under different humidity conditions. Finally, the input–output method is used to accurately evaluate the carbon reduction after adopting the optimal thickness under the condition of environmental humidity. See Figure 1 for details. By analyzing the research results of this paper, building engineers will be able to rationally design and choose the thickness of the building insulation materials which is more accurate and fits the actual operation.

**Table 3.** Research status of optimum thickness.

| Researcher | Research Contents | Research Results/Innovation |
|---|---|---|
| T.M.I. Mahlia, B.N. Taufiq, Ismail, H.H. Masjuki. [13] | This paper analyzes that the relationship between the thickness of building wall insulation materials. | The thermal conductivity is nonlinear and follows the polynomial function $x_{opt} = a + bk + ck^2$. |

**Table 3.** *Cont.*

| Researcher | Research Contents | Research Results/Innovation |
| --- | --- | --- |
| K. Comakli, B.Yuksel [14] | They studied the optimal insulation thickness of foamed polystyrene in the coldest city in Turkey and concluded that when the optimal thickness is used, obvious energy-saving effect can be achieved for cities with large number of humid days. | They proved the energy-saving effect of the optimal thickness and demonstrated the influence of the number of days on the optimal thickness and the payback period of investment. |
| A. Ucar, F. Balo [15] | The optimum insulation thickness of the external wall for four various cities from four climate zones of Turkey, energy savings over a lifetime of 10 years and payback periods are calculated for the five different energy types and four different insulation materials. | They proposed the optimal thickness corresponding relationship for different types of energy and different types of building insulation materials, and adopted P1-P2 method as the calculation method of the optimal thickness. |
| Huakun Huang, Yijun Zhou, Renda Huang, Huijun Wu, Yongjun Sun, Gongsheng Huang, Tao Xu. [16] | Taking the typical subtropical humid climate office building as the model, they established a full life cycle assessment model to calculate the optimal economic thickness of the new aerogel super insulation material d and further evaluated the energy saving rate, economic benefits, greenhouse gas emissions, etc. | They compared the building energy-saving effect caused by the optimal thickness of the new super thermal insulation material and the traditional building thermal insulation material and compared the energy saving rate, economic benefits, greenhouse gas emissions, etc., when using different building thermal insulation materials. |

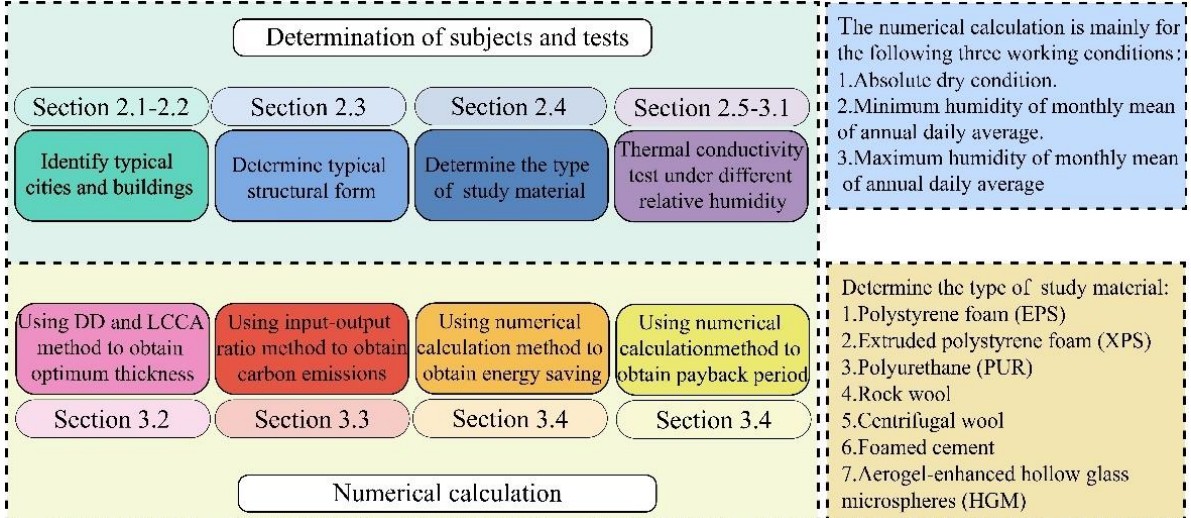

**Figure 1.** The research mind map.

## 2. Materials and Methodology

### 2.1. Typical Cities and Weather Data

In terms of land area, China covers about 1/15 of the world's landmass with a 9.73 million square kilometer territory. The vast land has resulted in the diversity of natural conditions, and various climates have been formed under the specific topographic and geomorphic characteristics. According to the climate characteristics, we divide China into severe cold area, cold area, hot summer and cold winter area, hot summer and warm winter area and mild area. In the study, according to the indicators in the code for thermal design of civil buildings [17], we selected ten typical cities with high relative humidity in five thermal zones as the research objects to study the optimal thickness change of thermal insulation materials under the influence of relative humidity and its environmental impact. Using the Xi'an University of Architecture and Technology's measured meteorological data from 1988 to 2017, we determined the cold and warm seasons of the different cities

and calculated the equivalent temperature difference and other basic data. See Table 4 for specific data of the cities.

**Table 4.** Basic data of ten typical cities.

| Thermal Zoning | Province | City | Maximum Humidity of Monthly Mean of Annual Daily Mean | Minimum Humidity of Monthly Mean of Annual Daily Mean | HDD (°C·d) | CDD (°C·d) | DD (°C·d) |
|---|---|---|---|---|---|---|---|
| Severe cold area | Heilongjiang | Tonghe | 80.65% | 59.65% | 5701 | 2 | 3001.22 |
|  | Jilin | Erdao | 81.10% | 55.77% | 5390 | 0 | 2836.84 |
| Cold area | Shandong | Chengshantou | 90.16% | 62.84% | 2698 | 2 | 1420.69 |
|  | Henan | Yucheng | 82.26% | 63.71% | 2306 | 99 | 1247.82 |
| Hot summer and cold winter area | Shaanxi | Hanzhong | 83.16% | 69.45% | 1920 | 68 | 1033.97 |
|  | Hubei | Enshi | 82.03% | 72.84% | 1541 | 98 | 844.85 |
| Hot summer and warm winter area | Fujian | Xiamen | 82% | 65.32% | 516 | 199 | 340.20 |
|  | Hainan | Haikou | 83.81% | 76.97% | 95 | 403 | 188.97 |
| Mild area | Guizhou | Zhijin | 81.90% | 73.87% | 1762 | 1 | 927.71 |
|  | Yunnan | Longling | 89.42% | 72.87% | 1285 | 0 | 676.32 |

## 2.2. The Typical Building Model

As the main body of climate, architecture is an important intermediate link in the process of analyzing the impact of climate on building materials. Therefore, a typical slab high-rise residential building in China was chosen and is modeled as a typical building in Table 5.

**Table 5.** Building model information.

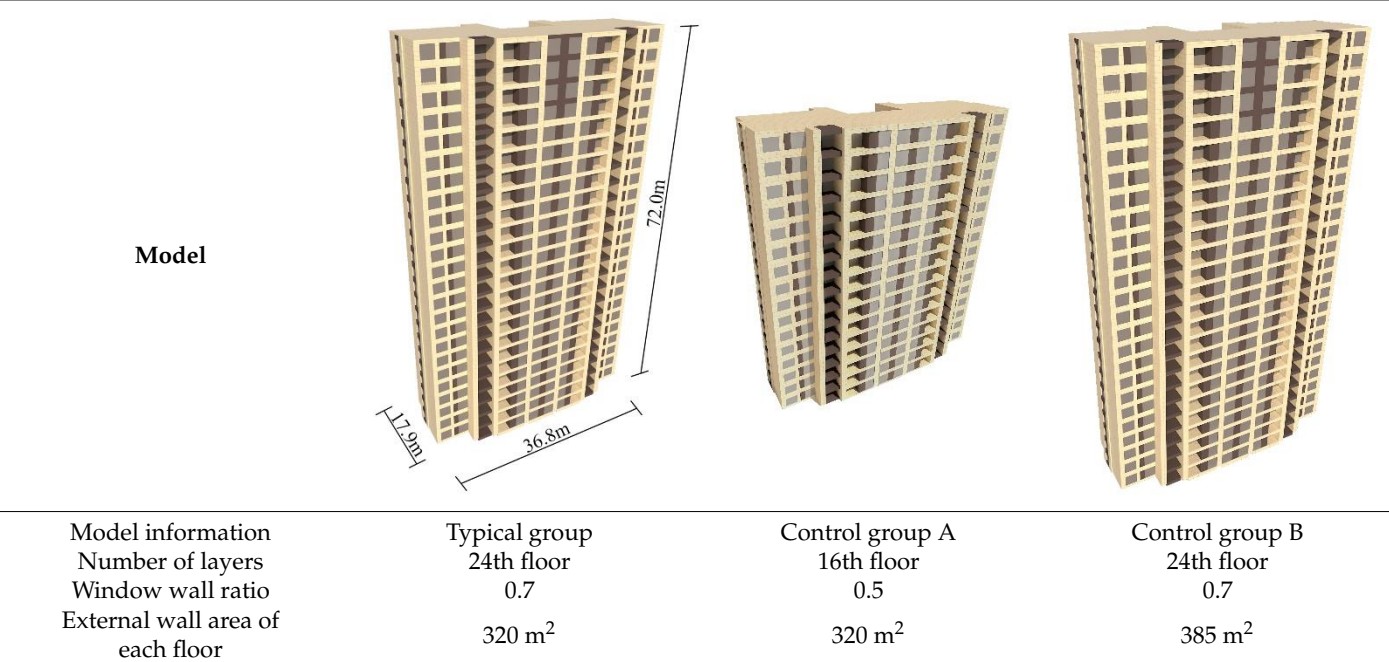

| Model information | Typical group | Control group A | Control group B |
|---|---|---|---|
| Number of layers | 24th floor | 16th floor | 24th floor |
| Window wall ratio | 0.7 | 0.5 | 0.7 |
| External wall area of each floor | 320 m$^2$ | 320 m$^2$ | 385 m$^2$ |

The typical building is a 24-story residential building with a length of 36.8 m, a width of 17.9 m and a height of 72 m. Each floor is composed of four apartment types and two elevators. The floor height of each floor is 3 m, and the building area is 530 m$^2$. The window wall ratio is 0.7, and the external wall area of each floor is 320 m$^2$. External walls are insulated using an external insulation structure. The roof adopts an inverted insulation form, and the floor between each floor does not have an insulation layer. The built-in partition wall is also separated by a light partition wall. In this study, another two control groups are set according to typical buildings to assess the environmental impact caused by the optimal thickness of the number of floors and the window wall ratio. The number of

floors of control group A is set as 16, the window wall ratio of control group B is 0.5, and the external wall of each floor is 385 m$^2$.

### 2.3. Construction

In the whole building envelope, the area of the external wall accounts for more than half, and the wall heat transfer energy consumption can reach about one-third of the total building energy consumption [18]. Building envelopes use heat transfer energy depending on the insulation material they are constructed of and the thickness of their insulation layers. According to different components of thermal insulation materials, the external wall insulation system is generally divided into a self-insulation system and a composite wall insulation system in the existing wall insulation system. The wall structure composed of a kind of building material with low thermal conductivity is called a self-insulation system. It can realize energy saving through the characteristics of building materials. Depending on where the insulation layer is located, there are three types of composite wall insulation: external wall external insulation system, external wall internal insulation system and sandwich insulation [19]. Compared with the four insulation systems, the exterior wall external insulation has strong applicability, which can effectively solve the heat loss problem of the weak links such as the thermal bridge in the external enclosure components and achieve the building energy-saving effect without affecting the indoor decoration and living conditions. Therefore, we chose the exterior wall external insulation system as the research object.

According to the national standard atlas external wall external insulation building structure (10J121) [20], A-type pasted insulation board external insulation system was selected as a typical structure. A-type pasted insulation board external insulation system is only mentioned in the standard drawings as applicable to EPS, XPS and PUR. However, it was learned from reading other documents [21,22] that A-type adhesive external insulation system is also applicable to adhesive external insulation materials such as rock wool, centrifugal cotton, aerogel-enhanced HGM and foamed cement. The structure of type A-pasted insulation board external insulation system in the standard drawings has a facing layer and a bonding layer, but it does not play a role in thermal insulation in the actual project. For the convenience of calculation, we simplified the structure and determined the specific thickness of each layer. The specific structure and thickness are shown in Figure 2 and Table 6.

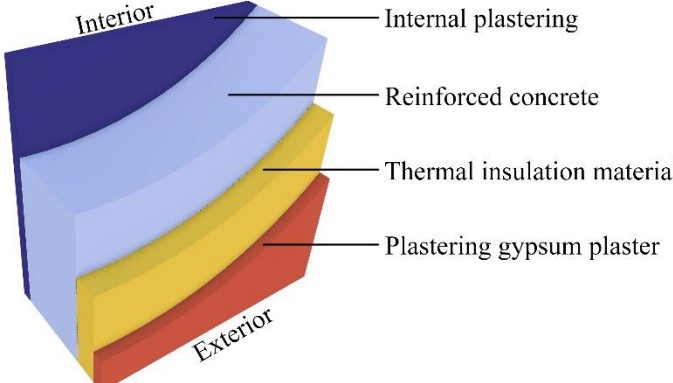

**Figure 2.** Simplified wall structure of A-type pasted insulation board external insulation system.

### 2.4. Test Materials

Building insulation materials can achieve low thermal conductivity in different forms, including loose filling, cotton wool or blanket, rigid form, in situ foaming or reflection form. Choosing the appropriate form and type of building insulation materials depends on the type of application and the physical, thermal and other properties of the materials required. However, traditional thermal insulation materials have become the preferred materials

in many building and thermal energy storage applications because of their low thermal conductivity and low cost [23]. However, with the continuous research on building thermal insulation materials, researchers have recently proposed a new type of building super thermal insulation material-aerogel-enhanced HGM, which is also taken as the research object in this study [24].

**Table 6.** Basic data of each layer of materials after simplification of A-type external insulation system.

| Material | Thickness (mm) | Thermal Conductivity (W/m·K) | Dry Density (kg/m$^3$) |
|---|---|---|---|
| Plastering gypsum plaster | 4 | 9.44 | 1500 |
| Thermal insulation material | $\delta_{im}$ | $k_{im}$ | $\rho_{im}$ |
| Reinforced concrete | 300 | 1.74 | 2500 |
| Internal plastering | 25 | 0.81 | 1600 |

Thermal conductivity can be affected by temperature, water content and humidity according to existing literature and published studies [25]. However, few people have further demonstrated that the change of thermal conductivity caused by the change of relative humidity will have an impact on the optimal thickness of insulation layer and the environment caused by the change of optimal thickness. As a result, we chose XPS, EPS, PUR, rock wool, centrifugal cotton, aerogel-enhanced HGM and foamed cement as the research objects for analyzing.

*2.5. Instrument*

Thermal conductivity of building insulation materials must be measured through experiments in order to be accurate. Due to the different applicable materials, test accuracy and test range of the measurement method, it is very important to select the appropriate test method for accurately obtaining the required measurement data. We selected the test method of constant temperature and humidity chamber and transient plane source method. The constant temperature and humidity chamber was used to provide stable working conditions for the experiment. Transient plane source is responsible for accurate determination of thermal conductivity. According to the transient plane source method, we selected TPS2200, and its basic test parameters are shown in Table 7. The TPS test principle is based on the thermal field generated by the plane circular heat source in the theoretically infinite medium to measure the instantaneous temperature rise of the tested piece. The plane probe designed by this technology is a metal nickel sheet with double helix structure after etching, and is wrapped with a Kapton or mica insulation layer about 0.025 mm thick to measure the change of thermal field. During the test, the heating time and heating power were set by the monitoring software and executed by the test equipment, as shown in Figure 3.

**Table 7.** TPS2200 Basic test parameters.

| Apparatus | Model | Test Temp Rage | Test Sample Requirements | Manufacturer | Range | Sample Type |
|---|---|---|---|---|---|---|
| Hot Disk Thermal constant analyzer | TPS2200 | RT °C | The min thickness is 2 mm, the min diameter is 10 mm, and the max size is not limited | Hot Disk AB | Thermal Conductivity: 0.01–500 W/m·K Thermal Diffusivity: 0.01~300 mm$^2$/s Specific Heat: 0.01~5 MJ/m$^3$·K | Solid, liquid, powder, paste, foam etc. |

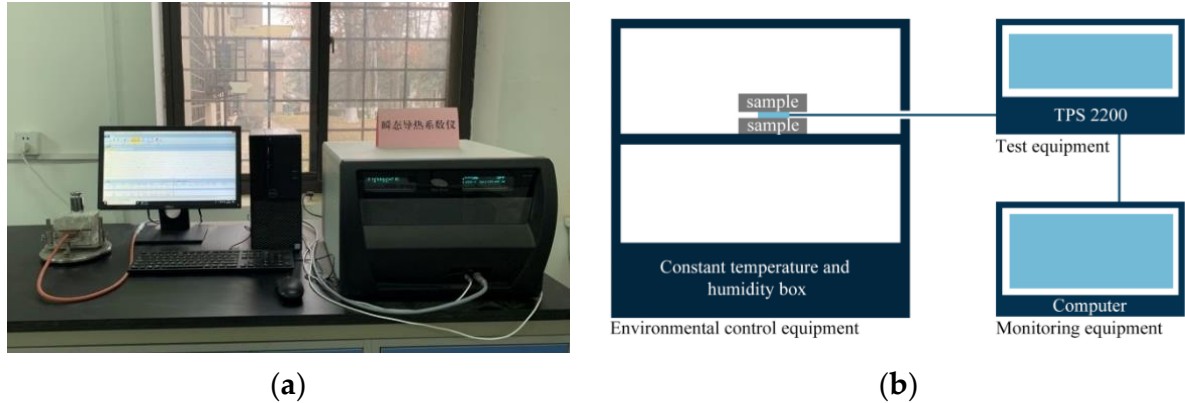

(**a**)                                                         (**b**)

**Figure 3.** TPS2200 test instrument: (**a**) physical drawing; (**b**) schematic diagram.

### 3. Results and Discussion

*3.1. Results of the Effect on Thermal Conductivity by Humidity*

The thermal conductivity of building insulation materials is significantly affected by environmental humidity [7,26]. A transient plane source and a constant temperature and humidity chamber were used to measure the thermal conductivity under different humidity conditions in this study. Finally, we obtain the functional relationship between the thermal conductivity and relative humidity of the seven insulation materials.

3.1.1. Experimental Process

A perfect experimental process will make the experimental results more accurate. After the research on the standards and previous papers, we determined the experimental process as follows:

1.  Test piece drying stage: Place the prepared test pieces in the drying oven. For materials that may undergo chemical changes or irreversible structural damage, 70 °C is used as the drying temperature. For materials that will not undergo chemical changes or irreversible structural damage, 105 °C is used as the drying temperature [27]. When measuring the mass change of the test piece over time at an interval of 24 h, it should not exceed 1% of its own mass three times in a row. The test piece is considered to have reached the constant weight.
2.  Sealing and cooling stage of test piece: Take the test piece out of the drying oven, wrap it with plastic film and place it indoors until it is cooled to room temperature.
3.  Test phase: After the sample is cooled, remove its plastic film and quickly measure its actual size and mass. Set the temperature of the constant temperature and humidity box to 25 °C and set up the humidity at 0%, 30%, 50%, 70%, 85% and 98% in succession. In the case of three consecutive weight measurements taken at intervals of 24 h, if the mass change of the test piece does not exceed 1%, the test piece has reached its constant weight. At this time, as the test pieces are placed up and down in pairs, the metal probe is placed in the middle of each and fixed, and then the thermal conductivity value is measured.
4.  Test repetition stage: Three groups of tests should be repeated for each insulation material a total of five times.

3.1.2. Experimental Data and Analysis

The thermal conductivity of building insulation materials was measured under the condition that the relative humidity was also changing gradually. Finally, the functional relationship between thermal conductivity and relative humidity was measured, as shown in Figure 3 and Table 8.

**Table 8.** Functional relationship between thermal conductivity and relative humidity of different materials.

| Material | Fitting Formula |
|---|---|
| EPS | $y = 1.5 \times 10^{-8}x^3 - 1.57 \times 10^{-6}x^2 + 5.66 \times 10^{-5}x + 0.02724$ |
| XPS | $y = 8.58 \times 10^{-9}x^3 - 1.24 \times 10^{-6}x^2 + 6.76 \times 10^{-5}x + 0.02963$ |
| Centrifugal cotton | $y = 3.01 \times 10^{-7}x^3 - 3.42 \times 10^{-5}x^2 + 9.54 \times 10^{-4}x + 0.02971$ |
| PUR | $y = 2.93 \times 10^{-8}x^3 - 4.43 \times 10^{-6}x^2 + 2.21 \times 10^{-8}x + 0.0445$ |
| Rock wool | $y = 2.67 \times 10^{-7}x^3 - 3 \times 10^{-5}x^2 + 8.48 \times 10^{-4}x + 0.05062$ |
| Aerogel-enhanced HGM | $y = 8.05 \times 10^{-8}x^3 - 1.41 \times 10^{-5}x^2 + 8.71 \times 10^{-4}x + 0.04789$ |
| Foam cement | $y = 2.16 \times 10^{-7}x^3 - 2.59 \times 10^{-5}x^2 + 0.00108x + 0.07134$ |

Figure 4 and Table 8 show the change of thermal conductivity of building insulation materials with the change of relative humidity. The fitting formula in Table 4 was used to accurately determine the corresponding thermal conductivity under different humidity. There was an almost linear relationship between thermal conductivity and relative humidity for EPS, XPS, PUR and aerogel-enhanced HGM. In environments with a relative humidity over 98%, changes in thermal conductivity of EPS, XPS and PUR are 16.8%, 9.4%, 13.3% and 52.1%, respectively. As the relative humidity began to rise, when it reached 30%, the thermal conductivity of foamed cement began to increase with the increase of relative humidity. Foamed cement undergoes a 16.7% change in thermal conductivity when relative humidity is 30%. Th thermal conductivity of foamed cement increases linearly with changing relative humidity when the relative humidity is increasing. In the presence of relative humidity above 70%, the thermal conductivity of foamed cement will rise sharply. When the relative humidity reaches 98%, the change of thermal conductivity is 85.4%. Thermal conductivity varies linearly and slowly with relative humidity before it reaches 70% for centrifugal cotton and rock wool. The thermal conductivity of centrifugal cotton and rock wool begins to increase sharply when the humidity reaches 70%, compared to the standard thermal conductivity under dry conditions. At 98% relative humidity, the variation in thermal conductivity is 167.9% and 95.3% respectively.

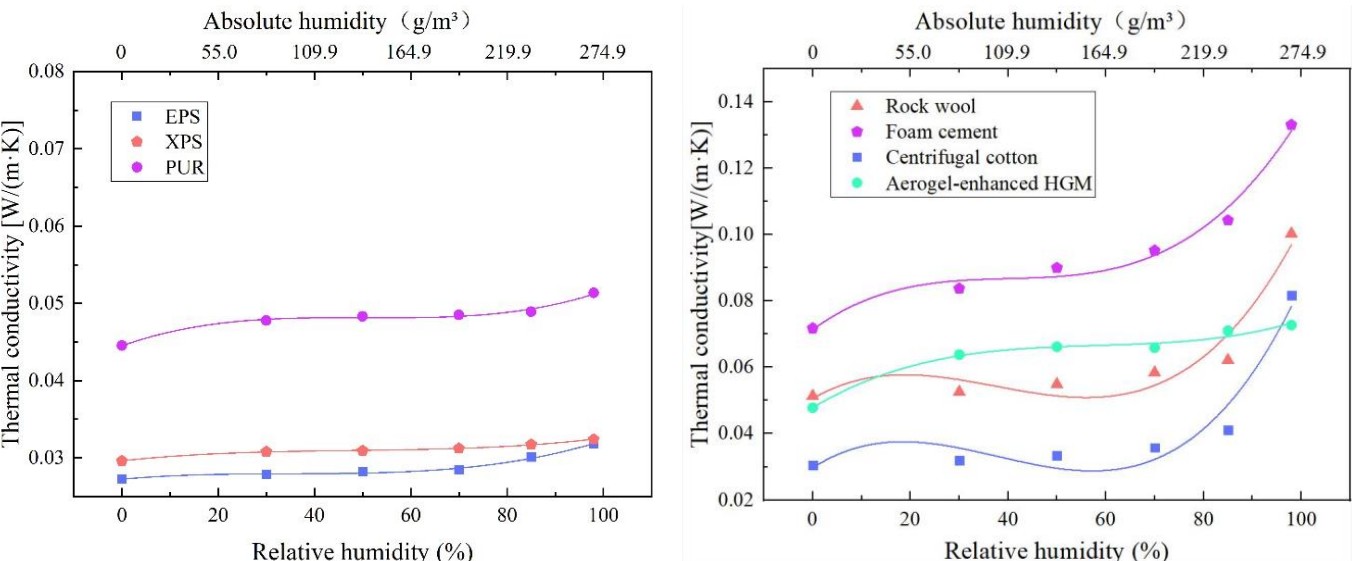

**Figure 4.** Functional relationship between thermal conductivity and relative humidity of different materials.

Thermal insulation materials will exhibit a great deal of difference in thermal conductivity based on the pore structure and porosity of its pores. EPS, XPS and other materials all have many small holes that are completely sealed, and XPS structures achieve closure rates of more than 99%. Such pore structure will make the wet air not shuttle well in the material

but can only stay in the closed pores on the surface, so that the thermal conductivity of a substance will not increase rapidly with changes in relative humidity. In contrast, foamed cement and other cement-based materials have strong water absorption capacity, and there are interconnected holes in the structure. Because the material itself has an adsorption capacity, the thermal resistance between solids decreases as relative humidity increases. A high relative humidity leads to a saturated state and liquid water begins to appear, which makes the thermal conductivity of the material increase sharply. Aerogel-reinforced materials are concrete materials before aerogel modification, and the concrete itself has interconnected cavities, so its thermal conductivity increases rapidly when the relative humidity just starts to increase. However, with the increase of thermal conductivity, its thermal conductivity does not increase significantly. This is because the aerogel itself has a strong hydrophobicity, so that its internal moisture does not increase much, resulting in such a result.

### 3.2. Results of the Optimum Thickness

In this study, the degree day method was used to calculate the heat transfer load of the steady-state enclosure structure, and then the LCCA was used to calculate its economic cost to solve the optimal thickness. For six different building insulation materials, the corresponding optimal insulation thickness of ten typical cities in China under three conditions was obtained, analyzed and compared. The relevant parameters are given in Tables 9 and 10.

**Table 9.** Performance of thermal insulation materials.

| Material | Dry Density (kg/m$^3$) | Price ($/m$^3$) |
|---|---|---|
| EPS | 16 | 56.51 |
| XPS | 4 | 117.74 |
| Centrifugal cotton | 23 | 75.35 |
| Rock wool | 200 | 91.05 |
| Foam cement | 374 | 59.65 |
| Aerogel-enhanced HGM | 260 | 493.88 |
| PUR | 40 | 204.08 |

**Table 10.** Parameters used in calculation.

| Parameter | Value |
|---|---|
| CDD and HDD | Table 4 |
| Related parameters in construction | Table 6 |
| $k_{im}$ | Figure 4 and Table 7 |
| $C_{im}$ | Table 9 |
| $\varphi$ | 0.90% |
| $\Phi$ | 3.70% |
| N | 20 years |
| $C_f$ | 0.1275 |
| $\mu_f$ | $3.6 \times 10^6$ |
| $\eta_{hs}$ | 0.99 |

### 3.2.1. Numerical Calculation Process

The heat loss of a building refers to the heat transfer and heat consumption through the enclosure structure and the air infiltration and air conditioning through the gaps of doors and windows. As the air infiltration through the gaps of doors and windows and the heat consumption of air conditioning are determined by the sealing performance and thermal insulation performance of the external windows, we will not consider it again. The insulation on the walls of the enclosure structure primarily determines the heat transfer and the heat consumption. Insulating a wall primarily involves choosing the right insulation

material and setting the thickness of the insulation sufficiently. The focus of this study is to consider the heat loss of the wall to optimize the thickness of the insulation layer.

The heat loss of the building envelope accounts for more than half of the total energy consumption of the building. According to the report of Hasan [28], the heat loss $Q$ (W/m$^2$) per unit area of the enclosure structure can be obtained by the following formula:

$$Q = U(T_i - T_{md}) \tag{1}$$

where $U$ (W/(m$^2$·K)) is the total heat transfer coefficient of the external wall, $T_i$ (°C) is the constant indoor comfortable temperature, and $T_{md}$ (°C) is the average daily temperature. The annual heat loss per unit area of external wall $Q_{an}$ (W/m$^2$) can be determined by the following formula:

$$Q_{an} = 86400 U DD \tag{2}$$

where degree day $DD$ (°C·d) is the function of heating degree day, air conditioning degree day and energy efficiency of heating and cooling system, which can be calculated by the following equation:

$$DD = \frac{CDD}{COP} + \frac{HDD}{\eta} \tag{3}$$

Annual energy demand $E_{an}$ (J/m$^2$-year) can be calculated by dividing the annual heat loss by the efficiency of the heating system $\eta_{hs}$ to determine, as follows:

$$E_{an} = \frac{Q_{an}}{\eta_{hs}} = \frac{86400 U DD}{\eta_{hs}} \tag{4}$$

The total heat transfer coefficient $U$ of the exterior wall structure is obtained from the reciprocal of the sum of the thermal resistances of each layer of the enclosure structure:

$$U = \frac{1}{R_{iaf} + R_{oaf} + R_w + R_{im}} \tag{5}$$

where $R_{iaf}$ ((m$^2$·K)/W) and $R_{oaf}$ ((m$^2$ K)/W) is the thermal resistance of inner and outer air layers, $R_w$ ((m$^2$·K)/W) is the total thermal resistance of all structural layers in the external wall except the insulation layer, and $R_{im}$ ((m$^2$·K)/W) is the thermal resistance of the insulation layer. As the thermal resistance of other parts of the enclosure except for the insulation layer is very small and subject to little environmental impact, the impact of environmental temperature and humidity on other parts of the enclosure is not considered. $R_{iaf}$ and $R_{oaf}$ have not been considered in some studies of optimal thickness [29]. In this study, these resistance terms are considered, which is closer to the actual use. $R_{im}$ can be expressed as:

$$R_{im} = \frac{\delta_{im}}{k_{im}} \tag{6}$$

where $\delta_{im}$ (m) and $k_{im}$ (W/(m·K)) represent the insulation thickness and thermal conductivity of insulation materials, respectively. The annual energy demand can be calculated as follows:

$$E_{an} = \frac{86400 DD}{(R_{iaf} + R_{oaf} + R_W + \frac{\delta_{im}}{k_{im}})\eta_{hs}} \tag{7}$$

Annual fuel consumption $m_{af}$ (kg/m$^2$) can be calculated by dividing Equation (7) by $\mu_f$:

$$m_{af} = \frac{86400 DD}{(R_{iaf} + R_{oaf} + R_W + \frac{\delta_{im}}{k_{im}})\mu_f \eta_{hs}} \tag{8}$$

where $\mu_f$ is the lower calorific value of a given fuel, usually in J/kg, J/m$^3$, or J/kWh, depending on the fuel type.

In the process of determining the optimal thickness through financial analysis, there are generally two methods: investment recovery method and LCCA. The investment

recovery method is based on the time required to repay the initial investment and the cost of energy saving in the operation stage brought by the investment. The disadvantage of this simple analysis method is that it does not consider currency inflation, which is also a very important financial consideration [30]. However, insulation thickness of building exterior walls can also be optimized using LCCA [31]. Therefore, we used an LCCA analysis method in this study.

Annual cost per unit area $C_{an}$ ($) can be obtained from the following formula:

$$C_{an} = \frac{86400DD}{(R_{iaf} + R_{oaf} + R_W + \frac{\delta_{im}}{k_{im}})\mu_f\eta_{hs}}C_f \tag{9}$$

where $C_f$ is the fuel cost in $/kWh. The total cost of the whole life cycle is calculated through LCCA and converted to present value by multiplying the total cost by the present value factor (*PWF*), where the service life of the building is $N$ (year). *PWF* is the inflation rate ($\varphi$) and interest rate ($\Phi$) function of.

$$PWF = (\frac{1-\varphi}{1+\phi})\left[1 - (\frac{1+\varphi}{1+\phi})^N\right] \text{ If } (\phi \neq \varphi) \tag{10}$$

$$PWF = \frac{N}{1+\phi} = \frac{N}{1+\varphi} \text{ If } (\phi = \varphi) \tag{11}$$

Building insulation material cost $C_{in}$ ($) is the unit price of the material multiplied by the thickness as follows:

$$C_{in} = C_{im}\delta_{im} \tag{12}$$

where $C_{im}$ ($/m^3$) is the unit price of insulation material; $\delta_{im}$ (m) is the thickness of the insulation material. Total cost of building heat loss $C_{tot}$ ($) should be the heating cost plus the cooling cost as follows:

$$C_{tot} = \frac{86400C_fPWFDD}{(R_{iaf} + R_{oaf} + R_W + \frac{\delta_{im}}{k_{im}})\mu_f\eta_{hs}} + C_{im}\delta_{in} \tag{13}$$

$\delta_{im}$ is obtained by derivation of Equation (13), and then the optimal thickness of the insulation material is as follows:

$$\frac{dC_{tot}}{d\delta_{im}} = \frac{d}{d\delta_{im}}\left[\frac{86400C_fPWFDD}{(R_{iaf} + R_{oaf} + R_W + \frac{\delta_{im}}{k_{im}})\mu_f\eta_{hs}} + C_{im}\delta_{im}\right] = 0 \tag{14}$$

$$\delta_{im} = 293.938\sqrt{\frac{DDC_fPWFk_{im}}{C_{im}\mu_f\eta_{hs}}} - (R_{iaf} + R_{oaf} + R_w)k_{im} \tag{15}$$

### 3.2.2. Data and Analysis

In the thermal insulation design of the building's peripheral protective structure, there is no need for excessive thermal insulation, which is planned to increase the initial investment cost in exchange for the reduction of energy cost, and there can be no lack of thermal insulation, which is to exchange the increase of energy cost for the reduction of initial investment cost. Therefore, it is vital to determine the optimal insulation thickness accurately in order to minimize the economic cost [32].

By solving the optimal thickness of seven kinds of insulation materials under three conditions, the change of their optimal thickness under the influence of humidity was obtained. The calculation was carried out successively for the selected ten different cities. See Table 11 for the specific values in detail.

The optimal thickness of building thermal insulation materials for a specific city increases with increasing humidity, as shown in Table 11. This is because the heat consumption of buildings are certain, and the thermal conductivity of materials increases with the

increase of humidity. If you want to achieve a comfortable indoor thermal environment, you can enhance the thermal insulation performance of the enclosure structure through the thickness of the insulation layer. Building insulation material thickness will be smaller if relative humidity is not taken into account, which will lead to a higher operating cost of the life cycle.

**Table 11.** Optimum insulation thickness of different insulation materials in ten typical cities under three working conditions (mm).

| Material | Working Condition | City | | | | | | | | | |
|---|---|---|---|---|---|---|---|---|---|---|---|
| | | Severe Cold Area | | Cold Area | | Hot Summer and Cold Winter Area | | Hot Summer and Warm Winter Area | | Mild Area | |
| | | Tonghe | Erdao | Chengshantou | Yucheng | Hanzhong | Enshi | Xiamen | Haikou | Zhijin | Longling |
| EPS | Dry | 192.71 | 187.09 | 129.54 | 120.79 | 109.08 | 97.66 | 58.40 | 41.04 | 102.81 | 86.35 |
| | Minimum humidity | 195.97 | 189.93 | 131.91 | 123.05 | 111.57 | 100.20 | 59.45 | 42.10 | 105.60 | 88.56 |
| | Maximum humidity | 200.03 | 194.34 | 136.66 | 125.54 | 113.88 | 101.38 | 60.46 | 42.45 | 106.73 | 90.77 |
| XPS | Dry | 135.63 | 131.57 | 90.00 | 83.68 | 75.22 | 66.98 | 38.62 | 26.08 | 70.69 | 58.81 |
| | Minimum humidity | 138.65 | 134.41 | 91.97 | 85.50 | 76.94 | 68.53 | 39.34 | 26.53 | 72.38 | 60.14 |
| | Maximum humidity | 139.57 | 135.42 | 93.04 | 86.05 | 77.35 | 68.79 | 39.52 | 26.59 | 72.62 | 60.61 |
| PUR | Dry | 120.25 | 116.47 | 77.75 | 71.87 | 63.99 | 56.31 | 29.89 | 18.21 | 59.77 | 48.70 |
| | Minimum humidity | 124.36 | 120.42 | 80.20 | 74.08 | 65.95 | 58.00 | 30.43 | 18.28 | 61.63 | 50.07 |
| | Maximum humidity | 125.19 | 121.18 | 81.31 | 74.58 | 66.35 | 64.05 | 30.54 | 18.29 | 61.86 | 50.53 |
| Centrifugal cotton | Dry | 174.47 | 169.32 | 116.63 | 108.62 | 97.90 | 87.44 | 51.50 | 35.60 | 92.15 | 77.09 |
| | Minimum humidity | 170.19 | 164.99 | 114.77 | 107.26 | 100.01 | 92.01 | 51.20 | 38.33 | 97.98 | 81.03 |
| | Maximum humidity | 203.22 | 198.57 | 155.84 | 128.61 | 117.09 | 102.58 | 59.31 | 40.92 | 108.11 | 100.02 |
| Rock wool | Dry | 200.58 | 194.50 | 132.26 | 122.80 | 110.13 | 97.78 | 55.32 | 36.54 | 103.34 | 85.55 |
| | Minimum humidity | 200.16 | 193.74 | 132.66 | 123.39 | 112.68 | 101.58 | 55.70 | 37.88 | 114.07 | 88.76 |
| | Maximum humidity | 221.47 | 215.60 | 158.51 | 136.50 | 123.02 | 180.03 | 59.84 | 38.88 | 107.99 | 99.98 |
| Foam cement | Dry | 294.14 | 285.26 | 194.35 | 180.52 | 162.02 | 143.98 | 81.96 | 54.53 | 152.11 | 126.11 |
| | Minimum humidity | 324.50 | 313.08 | 214.39 | 199.21 | 180.70 | 161.67 | 88.71 | 58.79 | 171.59 | 141.04 |
| | Maximum humidity | 346.05 | 336.21 | 238.22 | 211.65 | 190.07 | 167.28 | 92.46 | 59.60 | 177.00 | 150.92 |
| Aerogel-enhanced HGM | Dry | 73.63 | 71.11 | 45.3 | 41.38 | 36.10 | 31.01 | 13.10 | 5.61 | 33.31 | 25.93 |
| | Minimum humidity | 83.22 | 80.17 | 49.83 | 45.21 | 39.00 | 32.99 | 12.14 | 2.78 | 35.74 | 26.97 |
| | Maximum humidity | 84.02 | 81.04 | 50.54 | 45.50 | 39.20 | 33.07 | 11.99 | 2.60 | 35.83 | 27.02 |

Taking Chengshantou, the coastal city with the highest humidity, as an example, it can be observed in Figure 5. that the thickness changes of EPS, XPS and PUR are relatively small, and the change rates is between 1.8–5.5%. The thickness changes of centrifugal cotton and rock wool are significantly different from EPS, XPS and PUR at high humidity, but there is little difference with the above three materials under the minimum humidity condition. Their change rates under the minimum humidity condition, the maximum humidity condition and the absolute dry condition are −1.6% and 33.4%, 0.3% and 19.8%, respectively. Aerogel-enhanced HGM shows great changes under the condition of the lowest humidity, and the change rate is 10.0%. The change rate of the highest humidity condition is very small. Foamed cement shows great difference compared with the other five materials under either the lowest humidity condition or the highest humidity condition. This is because their thermal conductivity will increase greatly when the humidity is 0–30% and when

the humidity exceeds 70%. Their change rates under the lowest humidity condition, the highest humidity condition and the absolute dry condition are 2.6% and 19.5%, respectively. The reason for this result is that when calculating the optimal insulation thickness for the same kind of thermal insulation material in the same building, the only variable is the thermal conductivity, while the minimum humidity condition and the maximum humidity condition in Chengshantou are 62.84% and 90.16%, respectively. Except for EPS, XPS, PUR, aerogel-enhanced HGM and other materials whose thermal conductivity changes almost linearly with the change of humidity, other materials will change before and after 70% of materials. This makes the optimal thickness of thermal insulation material show a strong correlation with the thermal conductivity. In Figure 6, we bring in typical buildings to discuss the impact of three materials with different moisture change laws on their economic costs under different working conditions when the thickness changes. By comparing EPS, rock wool and foamed cement, it can also be observed that EPS has the least influence under the change of humidity. Rock wool changes little in low humidity but greatly in high humidity. Foamed cement is always changing with the change of humidity, which is also in line with the previous description of these three materials.

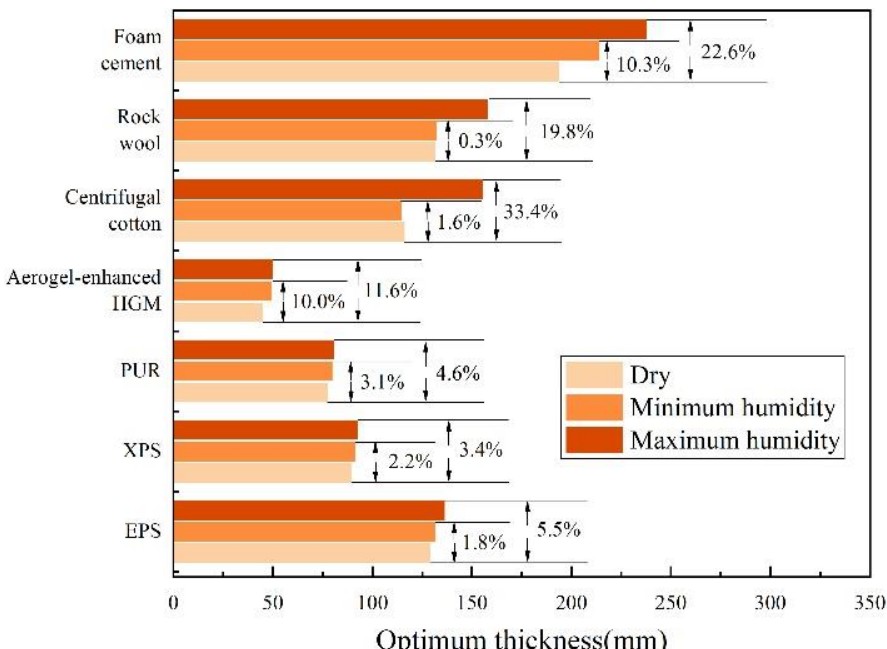

**Figure 5.** Optimum thickness of different thermal insulation materials under three working conditions in Chengshantou.

In Table 11, we can also observe that for different cities, the main factors affecting the optimal thickness of materials are DD value and the relative humidity of the city. Haikou and Hanzhong, which have little difference in maximum humidity but a large difference in DD value, are compared, as shown in Figure 7. The difference values of EPS and centrifugal cotton under absolute dry condition and maximum humidity condition are 165.8% and 168.2%, 175% and 186.1%, respectively. It can be seen that DD value is an indispensable and important parameter in the calculation of the optimal thickness. For Tonghe and Haikou with the largest difference in DD value, although their relative humidity is not exactly the same, under the absolute dry state, the optimal thickness difference values of EPS and centrifugal cotton are 369.6% and 344%, respectively. Therefore, the optimal thickness of each material in each city largely depends on the urban climate. On the basis of satisfying the urban climate, considering the influence of humidity change on the thermal conductivity will play a great role.

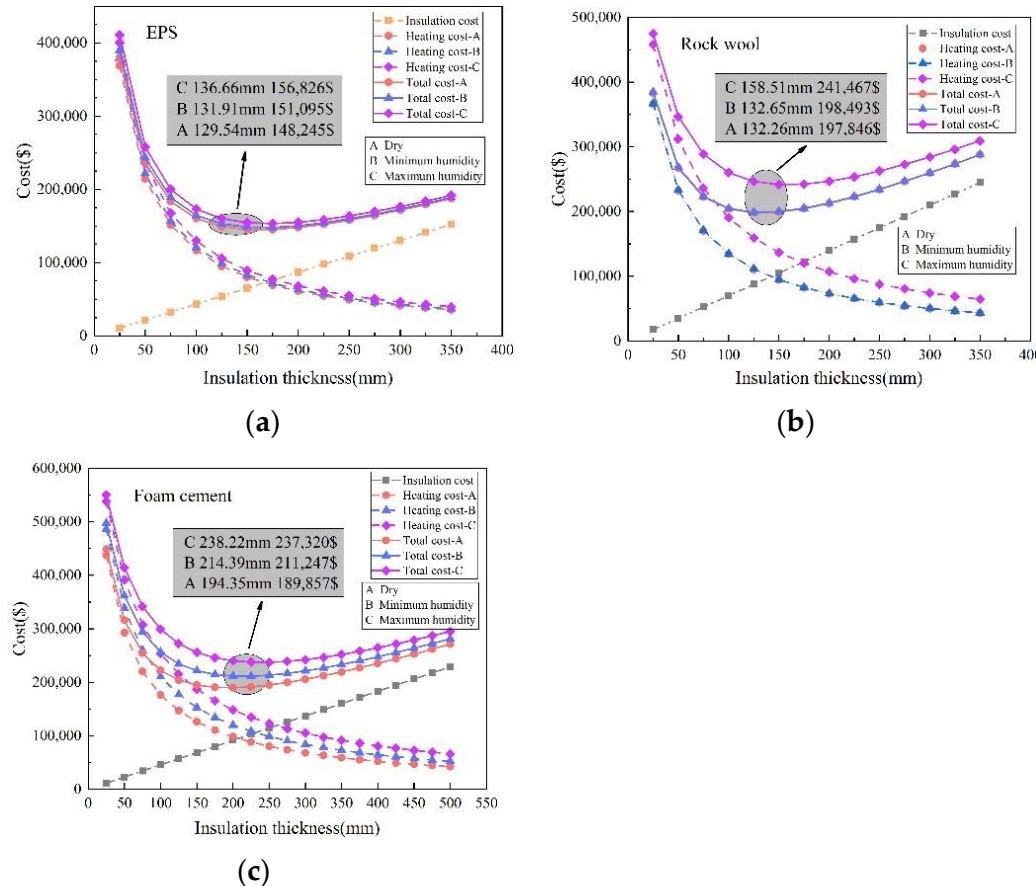

**Figure 6.** The influence of the thickness of different insulation materials on the economic cost of typical buildings in Chengshantou under three working conditions: (**a**) EPS; (**b**) rock wool; (**c**) foamed cement.

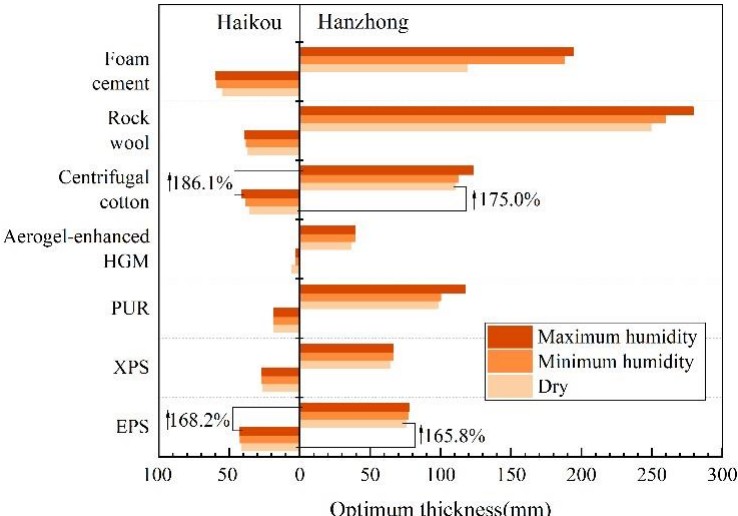

**Figure 7.** Comparison of the best thickness between Hanzhong and Haikou.

### 3.3. Results of the Greenhouse Gas Emissions

The optimal thickness of building insulation is used to reduce operating costs and greenhouse gas emissions, but changes in thermal conductivity will change the optimal thickness. However, the change of the optimal thickness will cause the change of carbon emissions [32]. In this paper, the input–output method is used to calculate the carbon

emissions of the optimal insulation thickness corresponding to seven different building insulation materials under three conditions in ten typical cities in China, and a comparative analysis is made.

### 3.3.1. Numerical Calculation Process

Building insulation materials and thickness settings serve a vital purpose in determining the efficiency of heating and cooling. The appropriate thickness setting and material selection will directly reduce the heating and cooling energy consumption, thus reducing carbon emissions and realizing building energy conservation. Among the existing carbon footprint analysis methods, the carbon emission factor method and the input–output method are commonly used. To calculate the annual carbon emissions, we used the input–output method to capture the relationship between energy efficiency and carbon emissions. The general equation for combustion is as follows [33,34]:

$$C_g H_y O_z S_w N_t + \alpha A(O_2 + 3.76 N_2) \rightarrow gCO_2 + \frac{y}{2}H_2O + wSO_2 + (\alpha - 1)AO_2 + BN_2 \quad (16)$$

The constants $g$, $y$, $z$, $w$ and $t$ in the formula are inconsistent for different fuel types. Considering China's economic situation, we take coal as the main reference, where $g$ = 7.078; $y$ = 5.149; $z$ = 0.517; $w$ = 0.01; and $t$ = 0.086. Constants $A$ and $B$ can be determined by element equilibrium:

$$A = (g + \frac{y}{4} + w - \frac{z}{2}) \quad (17)$$

$$B = 3.76\alpha(g + \frac{y}{4} + w - \frac{z}{2}) + \frac{t}{2} \quad (18)$$

The emission rate (*ER*) of combustion products produced by burning 1 kg fuel is calculated by the following formula:

$$ER_{CO_2} = \frac{gM_{CO_2}}{M_f} \equiv kgCO_2/kgfuel \quad (19)$$

$$ER_{SO_2} = \frac{wM_{SO_2}}{M_f} \equiv kgSO_2/kgfuel \quad (20)$$

where $M_f$ is the molecular weight of the fuel

$$M_f = 12g + y + 16z + 32w + 14t \quad (21)$$

$CO_2$ and $SO_2$ emissions can be calculated as follows:

$$m_{CO_2} = \frac{gM_{CO_2}}{M_f}m_{af} = \frac{44g}{M_f} \frac{86400DD}{(R_{iaf} + R_{oaf} + R_W + \frac{\delta_{im}}{k_{im}})\mu_f \eta_{hs}} \quad (22)$$

$$m_{SO_2} = \frac{wM_{SO_2}}{M_f}m_{af} = \frac{64w}{M_f} \frac{86400DD}{(R_{iaf} + R_{oaf} + R_W + \frac{\delta_{im}}{k_{im}})\mu_f \eta_{hs}} \quad (23)$$

### 3.3.2. Data and Analysis

The burning of fossil energy will produce greenhouse gases, such as $CO_2$ and $SO_2$, which will seriously damage the ecosystem. This has attracted widespread attention in China. Considering that most of China still uses coal-fired power generation, we bring the general equation constants of coal into the calculation. According to the observation of Formulas (22) and (23), the change rates of $CO_2$ and $SO_2$ emissions are consistent. We will not discuss them separately below.

For a specific city, the change law is obvious. Taking Chengshantou city as an example, the change rates of EPS, XPS and PUR insulation materials under the lowest and highest humidity conditions and absolute dry conditions are 2.0% and 6.0%, 2.5% and 3.8% and 4.0% and 5.9%, respectively from Figure 8. Centrifugal cotton and rock wool show a large

change rate at high humidity. Their change rates under the lowest humidity condition, the highest humidity condition and the absolute dry condition are −1.8% and 38.6% and 0.3% and 24.0%, respectively. Aerogel-enhanced HGM has a large change rate of 18.1% under the lowest humidity condition. This is because it is still in the stage of development completion, and its price is high. Therefore, its value is small when calculating the optimal thickness, which leads to the need for more fossil energy heating and cooling during the building operation stage. The change rate of foamed cement will be larger as a whole. The change rates of foamed cement under the lowest and highest humidity conditions and absolute dry conditions are 12.1% and 27.1%, respectively. Cement-based materials can be greatly reduced in greenhouse gas emissions by measuring their thermal conductivity precisely. We explore the influence of the thickness change of the insulation layer on the $SO_2$ emission of typical buildings for the three materials with different carbon emissions caused by the change of humidity, as shown in Figure 9. We can observe that $SO_2$ emissions gradually decrease with the increase of the thickness of thermal insulation materials because the increase of thickness will reduce the energy demand, thus reducing the combustion of fossil fuels. By comparing the influence of three materials on $SO_2$ emission with the change of thickness under three working conditions, the basic change law is basically consistent with the change law of thermal conductivity with humidity, realizing three different change laws. Through the horizontal comparison of six materials, it can be concluded that EPS is the least affected by humidity and the lowest carbon emission.

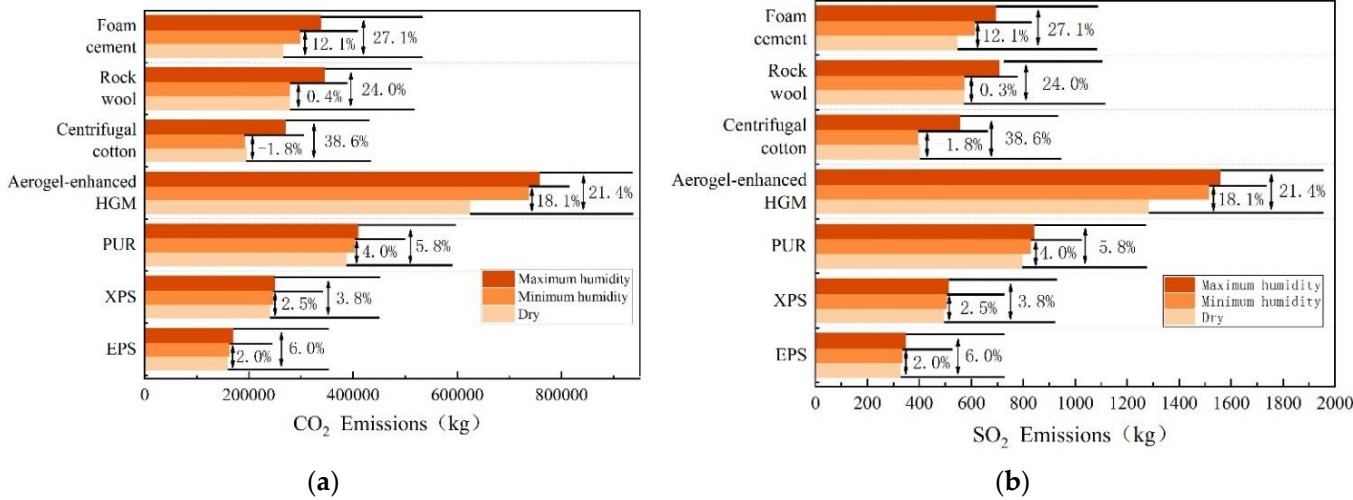

**Figure 8.** Carbon emissions of different materials under three working conditions: (**a**) $CO_2$ emissions; (**b**) $SO_2$ emission.

In the process of solving the optimal thickness of building thermal insulation materials, different results will not be caused by different building types, but greenhouse gas emissions will be caused by different building floors and window wall area ratios. We compared the $SO_2$ emissions of seven materials to three buildings, and the results are shown in Table 12. In the process of this algorithm, different layers will not lead to greenhouse gas emissions of each layer, so it is only the result of different layers. However, different window wall area ratios will lead to different exterior wall areas of each floor, leading to different greenhouse gas emissions of each floor, which will lead to different results of the whole building.

*3.4. Results of the Energy Saving and Payback Period*

3.4.1. Numerical Calculation Process

After determining the optimal insulation thickness under different relative humidity conditions, the energy saving under each condition can be calculated. Ozel [35] and Sisman [36] defined energy related savings as the difference between the total cost of a wall

without insulation and the total cost of an insulated wall with the best insulation thickness.

$$ES = \frac{86400DD}{(R_{iaf} + R_{oaf} + R_w)\mu_f\eta_{hs}}C_fPWF - C_{tot} \tag{24}$$

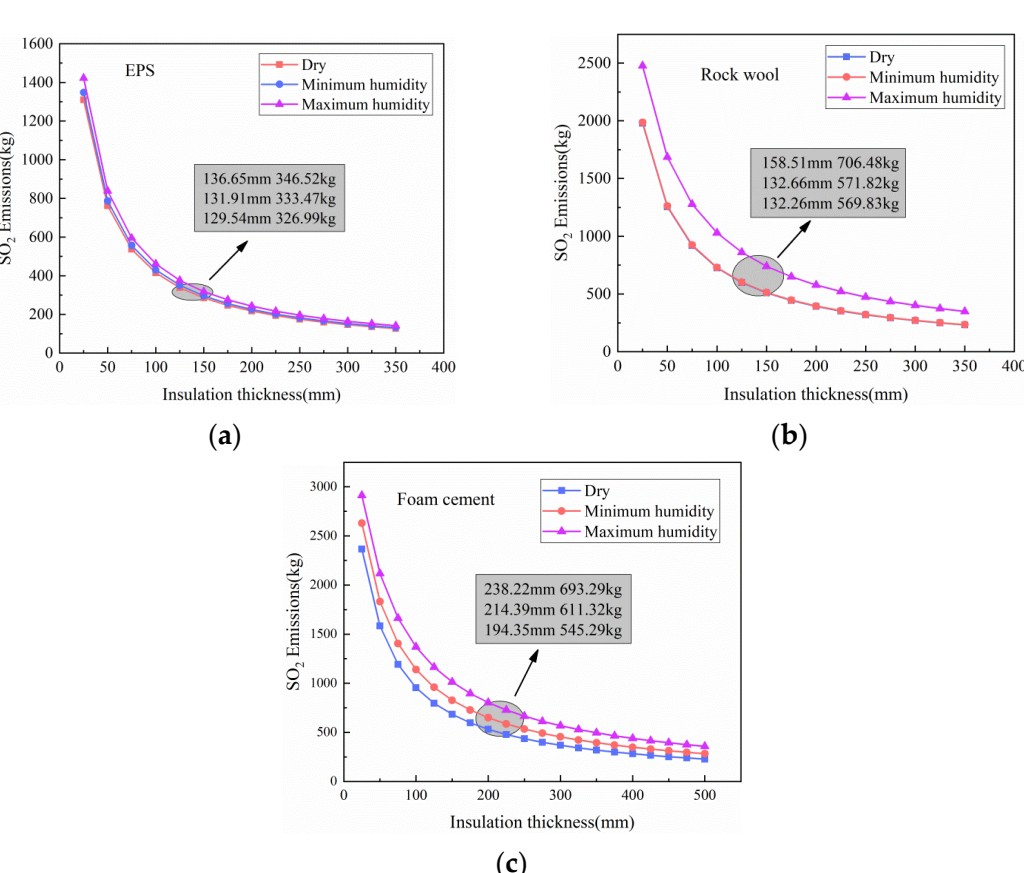

**Figure 9.** The influence of the thickness of thermal insulation materials of typical buildings in Chengshantou on SO₂ emission under different working conditions: (**a**) EPS; (**b**) rock wool; (**c**) foamed cement.

**Table 12.** Optimum insulation thickness of different insulation materials in ten typical cities under three working conditions (mm).

| Material | SO₂ Emission of the Whole Building | | | SO₂ Emission Per Layer | | |
|---|---|---|---|---|---|---|
| | Typical Group | Control Group A | Control group B | Typical Group | Control Group A | Control Group B |
| EPS | 327.2 | 218.1 | 393.6 | 13.6 | 13.6 | 16.4 |
| XPS | 492.3 | 328.2 | 592.3 | 20.5 | 20.5 | 24.7 |
| Centrifugal cotton | 399.4 | 266.2 | 480.5 | 16.6 | 16.6 | 20.0 |
| PUR | 794.1 | 529.4 | 955.4 | 33.1 | 33.1 | 39.8 |
| Rock wool | 569.9 | 379.9 | 685.6 | 23.7 | 23.7 | 28.6 |
| Aerogel-enhanced HGM | 1280.2 | 853.5 | 1540.2 | 53.3 | 53.3 | 64.2 |
| Foam cement | 545.3 | 363.5 | 656.0 | 22.7 | 22.7 | 27.3 |

Payback period is another important term in the economic analysis of the optimal insulation thickness. Payback period is defined as the length of time required to recover the investment cost. The payback period is one of the most important things to consider when deciding whether the project can be realized. The investment payback period covers a wide range because a longer payback period is usually undesirable, especially in the construction industry. Sisman [36] calculated the payback period (PP) of the investment in thermal insulation materials as follows:

$$PP = \frac{C_{in}}{ES}PWF \tag{25}$$

3.4.2. Data and Analysis

According to the results observed by Ozel [35] in the work on traditional thermal insulation materials, the saving effect increases with the increase of the thickness of thermal insulation materials and reaches the maximum value at the optimal thermal insulation thickness. Cuce pointed out that electricity is the most suitable energy in terms of energy conservation [32], so we use electricity as our calculated energy. Taking Chengshantou city as an example, Figure 10 shows that the energy-saving change rate of EPS, XPS and PUR building insulation materials is not large, which is basically in the range of 0.3–2.4%, while centrifugal cotton, rock wool, aerogel-enhanced HGM and foamed cement change greatly. The change rates of minimum humidity and maximum humidity and absolute dry working conditions are −0.3% and 7.1%, 0.1% and 6.6%, 13.3% and 15.6% and 3.2% and 7.1%, respectively. We explore the impact of the thickness change of the insulation layer on the energy efficiency of typical buildings for the three materials with different carbon emissions caused by the change of humidity. We can observe that with the increase of the thickness of the insulation material, the energy saving first increases and then decreases, and the maximum value is obtained at the optimal thickness. By comparing the influence of three materials on energy saving with the change of thickness under three working conditions, it can be seen that the basic change law is basically consistent with the change law of thermal conductivity with humidity, realizing three different change laws. Through horizontal comparison, EPS is the most energy-saving material among the seven materials.

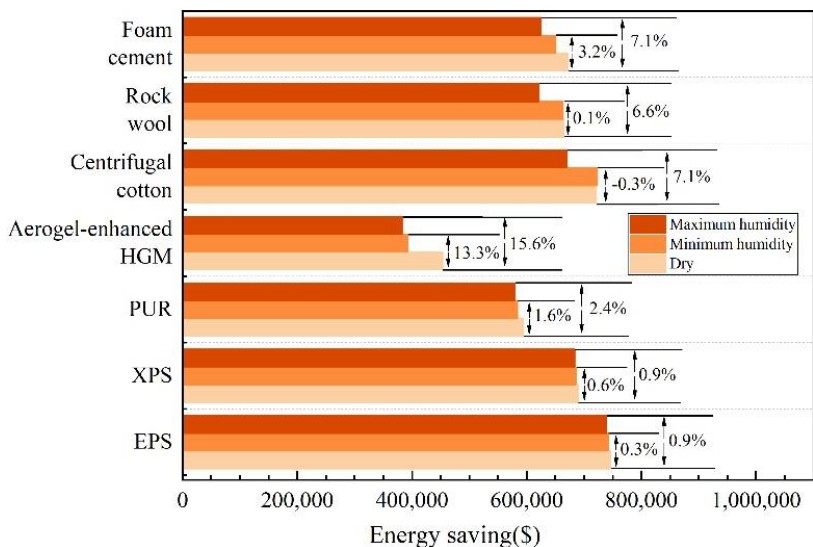

**Figure 10.** Energy saving of different materials of typical buildings in Chengshantou under three working conditions.

PP is not only related to the variables given in Formula (25), but also to the energy used. Cuce also points out that electricity is the most appropriate energy when considering the payback period because its energy saving is the highest [32]. Therefore, electricity was selected as the main energy source to judge the impact of relative humidity on the PP of building insulation materials. We still take Chengshantou city as an example. In Figure 11, with the increase of relative humidity, the PP value of the same material will also increase, but the change range is basically within half a year. According to Formula (25), PP is inversely proportional to energy saving, so the more energy-saving cities have, the shorter the payback period will be. Through horizontal comparison, EPS is the material with the shortest recovery period.

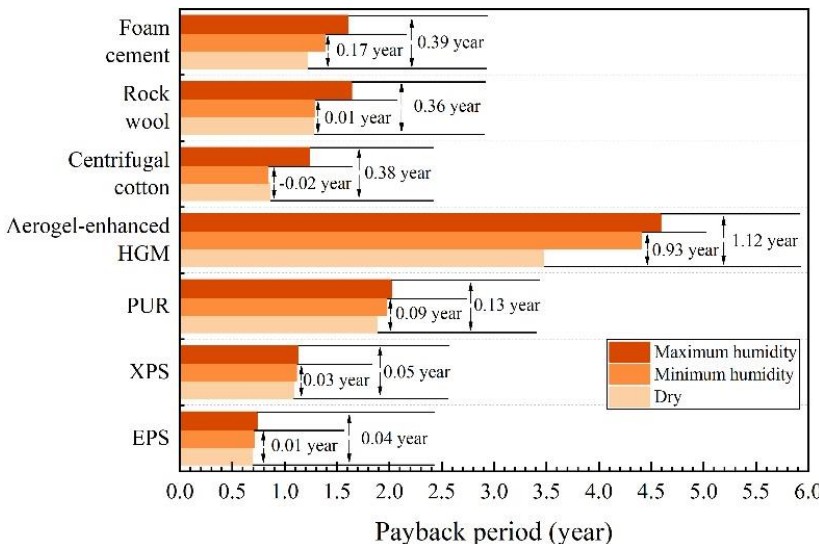

**Figure 11.** Recovery period of different materials under three working conditions in Chengshantou.

## 4. Conclusions

Residential buildings can improve their economic performance by using the best insulation thickness. In this study, considering the influence of relative humidity on the thermal conductivity of building insulation materials, an accurate method to determine the optimal thickness of insulation layer in external wall insulation considering the influence of relative humidity is proposed. This paper selects typical buildings in ten typical cities with high humidity in five thermal zones in China as the research object and adds two groups of contrast to explore the effect of shape coefficient and window wall area ratio. Taking the external wall insulation of typical residential buildings as an example, the optimal thickness of EPS, XPS, PUR, rock wool, centrifugal cotton, aerogel-enhanced HGM and foamed cement building insulation materials under absolute dry condition, minimum humidity condition and maximum humidity condition of monthly average of daily average over the years are calculated by degree day method and an LCCA economic model. Then, the carbon emissions under various working conditions are determined by the input–output ratio method. Finally, the recovery period and energy saving under various working conditions are obtained through calculation. The conclusion is that EPS is the material that is least affected by humidity, and when compared with other materials at the best thickness, the carbon emission is the smallest, the energy saving is the highest, and the recovery period is the shortest. The specific results are as follows:

1.  TPS technology was used to measure the relationship between seven materials and relative humidity. EPS, XPS, PUR and aerogel-enhanced HGM have a linear relationship between their thermal conductivity and relative humidity, while their growth is relatively flat. Centrifugal cotton and rock wool have linear thermal conductivities before relative humidity reaches 70%, the growth is relatively gentle but increases sharply after 70% relative humidity. The thermal conductivity of foamed cement increases twice when the relative humidity is 30% and 70%, and it is at a stable growth stage when the relative humidity is 30–70%. The change rates of thermal conductivity of EPS, XPS, PUR, centrifugal cotton, rock wool, aerogel-enhanced HGM and foamed cement under 98% relative humidity and absolute dry state are 16.8%, 9.4%, 13.3%, 167.9%, 95.3%, 52.1% and 85.4%, respectively.

2.  The optimum thickness of seven kinds of building insulation materials in ten typical cities was obtained by the degree day method and LCCA economic model analysis. The variation rates of the optimum thickness of EPS, XPS, PUR, centrifugal cotton, rock wool, aerogel-enhanced HGM and foamed cement in Chengshantou, a typical city with the highest humidity, under three working conditions are 0–5.5%, 0–3.4%,

0–4.6%, −1.6−33.4%, 0–19.8%, 0–11.6% and 0–22.6%, respectively. Compared with Haikou and Hanzhong, where the maximum relative humidity has little difference but the DD value has a large difference, the difference values of EPS and centrifugal cotton under absolute dry condition and maximum humidity condition are 165.8% and 168.2% and 175% and 186.1%, respectively. For Tonghe and Haikou with the largest difference in DD value, the optimal thickness difference of EPS and centrifugal cotton under absolute dry state is 369.6% and 344.0%, respectively.

3. The carbon emissions of seven building insulation materials in ten typical cities under different working conditions were obtained by using the input–output method. The carbon emission change rates of EPS, XPS, PUR, centrifugal cotton, rock wool, aerogel-enhanced HGM and foamed cement of Chengshantou, a typical city with the highest humidity, under three working conditions were 0–6.0%, 0–3.8%, 0–5.9%, −1.8–38.6%, 0–24.0%, 0–21.4% and 0–27.1%. By comparing the emissions between different cities with the continuous growth of DD value in cities, the carbon emissions of materials are also gradually increasing.

4. An energy saving and payback period can be determined based on the optimal thickness under various operating conditions. Among them, the typical city Chengshantou with the highest humidity has a small energy saving change rate of 0–2.4% under the three working conditions of EPS, XPS and PUR. However, the energy-saving change rates of centrifugal cotton, rock wool, aerogel-enhanced HGM and foamed cement under the three working conditions are large, which are −0.3–7.1%, 0–6.6%, 0–15.6% and 0–7.1%, respectively. However, except for aerogel reinforced materials, the recovery period of other materials increased within half a year.

5. To sum up, by comparing the thermal conductivity, optimal thickness, carbon emission, energy saving and recovery period of the six materials under three working conditions, we can conclude: Humidity has little effect on the thermal conductivity, optimal thickness, carbon emission, energy saving and recovery period of EPS, XPS and PUR materials. However, humidity shows great differences between centrifugal cotton and rock wool. Under low humidity conditions, the thermal conductivity, optimal thickness, carbon emission, energy saving and recovery period of centrifugal cotton and rock wool reflect the smallest change rate, even almost unchanged. However, it shows a large rate of change under high humidity conditions. The change rate of humidity on thermal conductivity, optimal thickness, carbon emissions, energy saving and recovery period of aerogel reinforced materials varies greatly under low humidity conditions but does not change much under high humidity conditions. Foamed cement is constantly affected by humidity. With the gradual increase of humidity, the change rates of thermal conductivity, optimal thickness, carbon emission, energy saving and recovery period are larger in these seven materials. In addition, among the seven building insulation materials, EPS is the most ideal building material. Because it has the minimum change of carbon emission, energy saving and recovery period before and after humidity change, and compared with other materials, it has the minimum carbon emission and recovery period and the maximum energy saving.

**Author Contributions:** W.Y.: data curation, conceptualization, methodology, funding acquisition; G.Z.: writing—original draft preparation, writing—reviewing and editing; W.H.: conceptualization, supervision; J.L.: resources. All authors have read and agreed to the published version of the manuscript.

**Funding:** This research was funded by China Postdoctoral Science Foundation Project (grant number 2021M702551), the independent research and development project of the State Key Laboratory of green building in Western China (grant number LSZZ202201), the key projects of Shaanxi Provincial Department of Education (grant number 20JS080), and the major project of the National Natural Science Foundation of China (grant number 51590910).

**Data Availability Statement:** Not applicable.

**Acknowledgments:** This study is provided by Xi'an University of architecture and technology with reference to urban basic meteorological data.

**Conflicts of Interest:** The authors declare that they have no known competing financial interests or personal relationships that could have appeared to influence the work reported in this paper.

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
