# Peer review of "The Effect of Relative Humidity Dependent Thermal Conductivity on Building Insulation Layer Thickness Optimization"

_buildings, doi:10.3390/buildings12111864_

Round 1

Reviewer 1 Report

The manuscript investigated the influence of relative humidity on the thermal conductivity of six building insulation materials. Instead of the numerical calculation process, I am more interested in the influence of relative humidity itself, which lacks detailed analysis. Therefore, I would like to reconsider this interesting manuscript after major revisions. 

1.      The absolute humidity can vary a lot in different seasons, while the relative humidity may not reflect the moisture change accordingly. Why was relative humidity used instead of absolute humidity? In the real situation, the environment temperature is changing, while the test temperature in this manuscript was fixed at 25 degrees Celsius. So how can the readers design the insulation layer thickness in a different and constantly changing environment temperature? Could the authors please comment on this?

2.      It would be better to use the full name instead of the abbreviation in the Abstract in line 22. Also, please make sure that when the abbreviation appears for the first time, there should be a full name.

3.      Typos: There should be a space between the number and unit in line 20. The equation numbers should be re-arranged to make them consistent in the whole manuscript. Please check the layout of the reference according to this journal’s guidance.

Author Response

Responds to the reviewer’s comments:

  1. Response to comment: (The absolute humidity can vary a lot in different seasons, while the relative humidity may not reflect the moisture change accordingly. Why was relative humidity used instead of absolute humidity?)

Response:Thanks for your constructive comments and suggestions. Absolute humidity is an important parameter of air conditioning engineering. It clearly expresses the amount of water vapor contained in the unit volume of wet air. The absolute humidity content will vary greatly in different seasons. At the same time, relative humidity is widely used in building thermal design because it can directly explain the impact of humid air on human thermal comfort and the humidity of rooms and enclosures.

In the teaching material of architectural physics (Liu Jiaping, Architectural Physics [M], Fourth Edition, Beijing: China Building Industry Press, 2009), there is a comparative case about absolute humidity and relative humidity. It is mentioned in the article that when the absolute humidity of the two rooms is the same, but the temperature is different, the saturated vapor pressure of others is different, but the actual vapor pressure in the room is not different, so the relative humidity is very different, which makes the users feel very different in the room. Therefore, it can be seen that the relative humidity can better reflect the humidity condition of the enclosure structure. Based on this, we chose the relative humidity.

In addition, most of the parameters set during the test of building insulation materials are relative humidity, and the absolute humidity under specific working conditions cannot be directly obtained through the test instrument. Therefore, we calculated the absolute humidity under different relative humidity conditions at 25 ℃ according to the following formula.

After checking that PS=3167.7 Pa at 25 ℃, the absolute humidity of different relative humidity at 25 ℃ is calculated as shown in the following Table 1.

Table 1. Conversion relationship between relative humidity

and absolute humidity at 25 ℃

Temperature (℃)

Relative humidity(%)

Absolute humidity(g/m³ï¼‰

25

0

0.0

25

30

82.5

25

50

137.4

25

70

192.4

25

85

233.6

25

98

269.4

We have shown the values of absolute humidity and relative humidity at 25 ℃ in Figure 4. in the manuscript.

Figure 4. Functional relationship between thermal conductivity and relative humidity of different materials

  1. Response to comment: (In the real situation, the environment temperature is changing, while the test temperature in this manuscript was fixed at 25 degrees Celsius. So how can the readers design the insulation layer thickness in a different and constantly changing environment temperature? Could the authors please comment on this?)

Response:Thanks for your constructive comments and suggestions. We are sorry that we did not explain in detail our weight analysis of the factors affecting the thermal conductivity by temperature and humidity.

In Table 2. of the manuscript, we mentioned the influence of temperature and humidity on building thermal insulation materials, and investigated the influence weight. Through the literature review, it is found that the influence of temperature on the change of thermal conductivity is far less than that caused by relative humidity. Therefore, this paper focuses on the influence of relative humidity on the thermal conductivity and weakens the influence of temperature on it.

During the test, this paper also conducted relevant tests on the influence of temperature on the thermal conductivity, measured the thermal conductivity of the above seven building thermal insulation materials using the GHP 456 protective hot plate method, and processed and fitted the test results. The results are shown in Table 2. and Figure 1. The change rates of thermal conductivity of EPS, XPS, PUR, rock wool, centrifugal cotton, aerogel-enhanced HGM and foamed cement when the temperature changes from 20 ℃ to 70 ℃ are 23.7%, 25.6%, 32.2%, 20.1%, 29.9%,11.7% and 16.2% respectively, while the change rates of thermal conductivity when the relative humidity changes from 0% to 98% are 16.8%, 9.4%, 13.3%, 95.3%, 167.9%, 52.1% and 85.4% respectively. It can be observed that in addition to foam based hydrophobic building insulation materials such as EPS, XPS, PUR, etc., humidity has a greater impact on the thermal conductivity than temperature change, even reaching more than five times.

Table 2. The functional relationship between thermal conductivity

and temperature of different materials

Material

Fitting formula

EPS

 y=1.68743E-4x+0.0338

XPS

 y=1.33543E-4x+0.0227

Centrifugal cotton

 y=5.20857E-4x+0.07759

PUR

 y=1.60514E-4x+0.02283

Rock wool

 y=1.30514E-4x+0.03166

Aerogel-enhanced HGM

 y=1.07866E-4x+0.04728

Foam cement

 y=2.04829E-4x+0.06221

Figure 1. The functional relationship between thermal conductivity

and temperature of different materials

  1. Response to comment: (It would be better to use the full name instead of the abbreviation in the Abstract in line 22. Also, please make sure that when the abbreviation appears for the first time, there should be a full name.)

Response:Thanks for your constructive comments and suggestions. We have changed the abbreviation of the summary in line 22 to the full name, and checked the full text in detail where the abbreviation appears.

  1. Response to comment: (There should be a space between the number and unit in line 20.)

Response:Thanks for your constructive comments and suggestions. We have changed the space between numbers and units in line 20 you raised, and checked in detail whether the full text has the same problem, and corrected it.

  1. Response to comment: (Please check the layout of the reference according to this journal’s guidance)

Response:Thanks for your constructive comments and suggestions. We carefully read the format requirements of this magazine and corrected our references one by one according to the requirements.

Reviewer 2 Report

The article is an interesting article that provides very useful information for scientists and technicians in the field of building renovation.

It is quite a long article and sometimes there are repetitions that could be avoided / eliminated and therefore it could be a bit shortened in length.

Author Response

Responds to the reviewer’s comments:

  1. Response to comment: (It is quite a long article and sometimes there are repetitions that could be avoided / eliminated and therefore it could be a bit shortened in length.)

Response:Thanks for your constructive comments and suggestions. We express the research status of the optimal thickness and the influencing factors of building thermal insulation materials in the introduction in the form of a table, thus shortening the length of the article. We have deleted the introduction on the background of building thermal insulation materials in Section 2.4, so as to achieve a more concise purpose. In addition, we have also deleted Chapter 3. During our analysis, we found that the laws of greenhouse gas emissions, energy conservation and humidity changes in the recovery period of buildings are consistent with those of the optimal thickness of building thermal insulation materials. Therefore, we have weakened the analysis process of 3.3 and 3.4 in this revision.

Reviewer 3 Report

The work is of high relevance, any effort to optimize energy consumption must be considered with the current problems. The work is very well presented from a conceptual and developmental point of view. The only observation I ask is to include a reference that justifies that the humidity ranges evaluated (for China), correspond to the generality of the ranges found in the world. Very good work.

Author Response

Responds to the reviewer’s comments:

  1. Response to comment: (The only observation I ask is to include a reference that justifies that the humidity ranges evaluated (for China), correspond to the generality of the ranges found in the world.)

Figure 1. Distribution of global climate zones based on the average near surface temperature (T) and relative humidity (RH) observed between 1961 and 1990

Response:Thanks for your constructive comments and suggestions. A. J. Teuling et al. (Teuling, A. J., et al. Bivariate Colour Maps for Visualizing Climate Data. International Journal of Climatology 2022, 31, 1408–1412) proposed a color representation method based on two-dimensional color legend and divergent data to make global data visual. Figure 1 is the global temperature and humidity map drawn by this method. It can be seen from the figure that both dark purple and light blue areas are the colors when the relative humidity reaches 80%. It can be seen that more than two-thirds of the regions with high humidity in the world can be seen. It is of great practical significance to consider the impact of relative humidity on the thermal conductivity in China and even in the world.

Reviewer 4 Report

1. The objective of the proposed work does not contribute anything with respect to the published literature. The authors should improve this aspect. I suggest that you make a review table of recent works and comment on what these works contribute and the gap that they are going to fill.

2. Congratulations to the authors for figure 1. It makes the proposed work very clear. I think the humidity is not an interesting or adequate result. Humidity has to do with the tightness of the building, not with the level of insulation. Hermeticity equal to gaps in the envelope (window frames, grids, windows, doors...) all this is neither commented nor treated.

3. The authors must analyze the work. The modification they make to the building with the insulation should not generate changes in humidity. Explain this better to make a decision.

In order to continue reviewing the article, the authors are required to analyze the previous points well. The authors mix results of the building with results of the materials. The abstract comments building results. They should define a sample of buildings, a sample of different typologies, geometries, % glass, etc... different uses of the building and even the different climates that they propose. With all this, comment on the differences of insulating inside, outside or in the middle of the wall. After the above, to show why they do laboratory experiments on commercial materials that do not require innovation. No high conductivity insulation or any new material has been proposed. And finally review all the established formulation. The U-value of the building is not what they present in their formulas. There are more things like thermal bridges.

The article in its current state should be rejected, unless they make a great effort to modify it.

Author Response

Responds to the reviewer’s comments:

  1. Response to comment: (I suggest that you make a review table of recent works and comment on what these works contribute and the gap that they are going to fill.)

Response:Thanks for your constructive comments and suggestions. According to your comments, we have made a literature analysis for each part and summarized their respective contributions. See Table 1.2.3. in the manuscript for details. We point out the lack of current research and propose our innovation points: Based on the above research on the status quo, almost all the current researches on the optimal thickness ignore the influence of environmental humidity on the thermal conductivity. The thermal conductivity is based on the measured thermal conductivity of the material in the absolute dry state. This will cause the design value of thermal conductivity to be lower than the actual value, resulting in insufficient design of insulation thickness, affecting users' thermal comfort and increasing building energy consumption. Through the research on the influence of temperature and humidity on building thermal insulation materials, it is found that the humidity driven change of thermal conductivity is significantly greater than the temperature driven change. Therefore, in practical engineering applications, we should first consider the influence of humidity when selecting the thickness of building insulation materials, especially when building walls are exposed to extreme humidity for a long time, so as to better realize building energy conservation and reduce greenhouse gas emissions.

Table 1. Research Status of Building Thermal Insulation Materials Affected by Ambient Temperature and Humidity

Researcher

Research contents

Research results/Innovation

I. Budaiwi,

A. Abdou.[6]

The impact of the k-value change of fibrous insulation materials (i.e. fiberglass) in a typical wall–roof system due to moisture content levels on the thermal and energy performance of a typical residential building under hot–humid climatic conditions is investigated.

Moisture performance is investigated utilizing theoretical longterm hygrothermal performance modeling and simulation techniques. Layer-and time-averaged levels of moisture content in the fibrous insulation are determined and the corresponding k-value change is evaluated from measured relationships. The impact of the k-value change due to moisture on the building thermal load and cooling energy performance of a residential building is then assessed utilizing detailed building energy simulation software.

K.J. Kontoleon, C. Giarma.[7]

This paper investigates the impact of moisture content on the thermal inertia parameters of building material layers. Their consideration is essential to enhance the design of building elements, from a thermal point of view, when exposed to varying moisture content conditions.

Moisture content and relative humidity variations of each analysed layer, as defined by specific moisture storage functions, are shown to interrelate non-linearly with the layer resistor–capacitor circuit section parameters (thermal conductivity and volumetric heat capacity) showing notable consequences on the thermal inertia parameters.

Table 2. Research Status of Building Thermal Insulation Materials Affected by Temperature and Humidity

Researcher

Research contents

Research results/Innovation

I. Budaiwi,

A. Abdou,

M. Al-Homoud.

[8]

They revealed the relationship between the temperature and thermal conductivity of various locally produced insulating materials.

The impact of thermal conductivity variation with temperature on the envelope-induced cooling load for a theoretically modeled building is quantified and discussed.

M. Khoukhi,

M. Tahat.[9]

This study is to investigate the relationship between the temperature and thermal conductivity of various densities of polystyrene, which is widely used as building insulation material in Oman.

The impact of thermal conductivity variation with temperature on the envelope-induced cooling load for a simple building model is discussed.

Hoseini,

Atiyeh,

Majid Bahrami.

[10]

This work presents a comprehensive investigation of aerogel blankets thermal conductivity (k-value) in humid conditions at transient and steady state regimes. Transient plane source (TPS) tests revealed that the k-value of aerogel blankets can increase by up to approximately 15% as the ambient relative humidity (RH) increases from 0% to 90% at 25 °C.

This paper mechanisms affecting the k-value of aerogel blankets as a function of RH and T are investigated.

Alvey,

Jedediah B,

Jignesh Patel,

Larry D,

Stephenson.

[11]

In this paper, the thermal conductivity of several commercially available insulating materials (three kinds of aerogel composite blankets, two kinds of extruded polystyrene foam (XPS) and one kind of foamed polyurethane foam (PUR)) is evaluated as affected by ambient temperature and humidity.

Results indicate that humidity levels play a significant role in PUR performance, but not a significant role in XPS performance. The three aerogel composites have mixed results: one has little relationship between moisture content and thermal performance, one is strongly affected by moisture and the remaining is moderately affected by moisture.

Nosrati,

Roya Hamideh,

Umberto Berardi.

[12]

In this paper, the change of thermal conductivity of aerogel reinforced materials in the temperature range of − 20 ℃to +60 ℃ and the relative humidity (RH) range of 0% to 95% is studied.

This study shows that compared to the standard testing condition, the maximum increase in the thermal conductivity was 100% under 95% RH, while the greatest temperature-driven increase in the thermal conductivity was 12% at the maximum tested temperature. The humidity-driven changes in the thermal conductivity of aerogel-based products are significantly greater than temperature-driven changes.

Table 3. Research status of optimum thickness

Researcher

Research contents

Research results/Innovation

T.M.I. Mahlia,

B.N. Taufiq,

Ismail,

H.H. Masjuki.[13]

This paper analyzes that the relationship between the thickness of building wall insulation materials.

The thermal conductivity is nonlinear and follows the polynomial function xopt = a + bk + ck2.

K. Comakli,

B.Yuksel[14]

They studied the optimal insulation thickness of foamed polystyrene in the coldest city in Türkiye, and concluded that when the optimal thickness is used, obvious energy-saving effect can be achieved for cities with large number of days.

They proved the energy-saving effect of the optimal thickness, and demonstrated the influence of the number of days on the optimal thickness and the payback period of investment.

A. Ucar,

F. Balo[15]

The optimum insulation thickness of the external wall for four various cities from four climate zones of Turkey, energy savings over a lifetime of 10 years and payback periods are calculated for the five different energy types and four different insulation materials.

They proposed the optimal thickness corresponding relationship for different types of energy and different types of building insulation materials, and adopted P1-P2 method as the calculation method of the optimal thickness.

Huakun Huang,

Yijun Zhou,

Renda Huang,

Huijun Wu,

Yongjun Sun,

Gongsheng Huang,

Tao Xu.[16]

Taking the typical subtropical humid climate office building as the model, they established a full life cycle assessment model to calculate the optimal economic thickness of the new aerogel super insulation material d, and further evaluated the energy saving rate, economic benefits, greenhouse gas emissions, etc.

They compared the building energy saving effect caused by the optimal thickness of the new super thermal insulation material and the traditional building thermal insulation material, and compared the energy saving rate, economic benefits, greenhouse gas emissions, etc. when using different building thermal insulation materials.

  1. Response to comment: (Humidity has to do with the tightness of the building, not with the level of insulation. Hermeticity equal to gaps in the envelope (window frames, grids, windows, doors...) all this is neither commented nor treated.)

Response:Thanks for your constructive comments and suggestions. We have added the effect of sealing on relative humidity in line 100-105: Although in some areas with high humidity, the construction of buildings may consider the installation of moisture-proof and reduce the moisture components inside the envelope, but in the production, transportation and installation of building materials, the humidity inside the building insulation materials will also reach a dynamic balance with the surrounding environment, so it is important to consider the impact of humidity on the building.

  1. Response to comment: (The modification they make to the building with the insulation should not generate changes in humidity. Explain this better to make a decision.)

Response:Thanks for your constructive comments and suggestions. In the process of reproduction, transportation and construction of building thermal insulation materials, it has always been in the process of dynamic balance with the environment under the local humidity conditions, while the data selected in the previous calculation of the optimal thickness are measured in the absolute dry state, which is not consistent with the actual situation. When considering the optimal thickness of building thermal insulation materials, the humidity factor is considered because it has reached the humidity condition of the environment itself before installation, rather than being in an absolute dry state. We mainly propose this paper for this point.

  1. Response to comment: (They should define a sample of buildings, a sample of different typologies, geometries, % glass, etc... different uses of the building and even the different climates that they propose. With all this, comment on the differences of insulating inside, outside or in the middle of the wall.)

Response:Thanks for your constructive comments and suggestions. On the basis of the original typical buildings, we propose two other buildings for the typical buildings. See Table 5. in the manuscript for details. One is to change the number of floors of the building, and the other is the window wall ratio of the building, so as to compare the impact of the optimal thickness on different building types. On the basis of this, we also compared the carbon emissions of the three buildings under the optimal thickness of humidity, and drew Table 12. in the manuscript.

Table 5. Building Model Information

Model

Model information

Typical group

Control group A

Control group B

Number of layers

24th floor

16th floor

24th floor

Window wall ratio

0.7

0.5

0.7

External wall area of each floor

320 m2

320 m2

385 m2

Table 12. Optimum insulation thickness of different insulation materials in ten typical cities under three working conditions (mm)

SO2 emission of the whole building

SO2 emission per layer

Material

Typical group

Control group A

Control group B

Typical group

Control group A

Control group B

EPS

327.2

218.1

393.6

13.6

13.6

16.4

XPS

492.3

328.2

592.3

20.5

20.5

24.7

Centrifugal cotton

399.4

266.2

480.5

16.6

16.6

20.0

PUR

794.1

529.4

955.4

33.1

33.1

39.8

Rock wool

569.9

379.9

685.6

23.7

23.7

28.6

Aerogel-enhanced HGM

1280.2

853.5

1540.2

53.3

53.3

64.2

Foam cement

545.3

363.5

656.0

22.7

22.7

27.3

  1. Response to comment: (After the above, to show why they do laboratory experiments on commercial materials that do not require innovation. No high conductivity insulation or any new material has been proposed.)

Response:Thanks for your constructive comments and suggestions. Traditional building thermal insulation materials are still widely used in China's current housing construction process, so we have selected six traditional building thermal insulation materials. We listened to your opinions and added new aerogel materials in this article.

  1. Response to comment: (The U-value of the building is not what they present in their formulas. There are more things like thermal bridges)

Response:Thanks for your constructive comments and suggestions. The heat loss of the enclosure structure calculated in our calculation is mainly for the flat wall of the enclosure structure, which does not include the weak links in the enclosure structure such as thermal bridges. This method is often used in the calculation of the optimal thickness of the existing enclosure structure, but your comments are very good. We will try to solve this problem in the next work and propose a more accurate calculation method.

Round 2

Reviewer 1 Report

I think the manuscript is acceptable now.

Reviewer 4 Report

The authors modified to improve the description of the methodology. The current version is better than the previous one. However, confusion is generated between the simulation tools used to evaluate buildings. This should not be in doubt because it is the most important point of the work. The novelty of the work is limited, but the results are well presented.